# Oscillations in an artificial neural network convert competing inputs into a temporal code

Katharina Duecker[1,2]*, Marco Idiart[3], Marcel van Gerven[4], Ole Jensen[1]

**1** Centre for Human Brain Health, School of Psychology, University of Birmingham, Birmingham, United Kingdom, **2** Department of Neuroscience, Brown University, Providence, Rhode Island, United States of America, **3** Institute of Physics, Federal University of Rio Grande do Sul, Porto Alegre, Brazil, **4** Donders Institute for Brain, Cognition and Behaviour, Radboud University, Nijmegen, the Netherlands

\* katharina.duecker@gmail.com

**Data Availability Statement:** The code is publicly available at https://github.com/katduecker/dnn_osci.

**Funding:** This work was supported by a Wellcome Trust Discovery Award (227420 to OJ) and the

## Abstract

The field of computer vision has long drawn inspiration from neuroscientific studies of the human and non-human primate visual system. The development of convolutional neural networks (CNNs), for example, was informed by the properties of simple and complex cells in early visual cortex. However, the computational relevance of oscillatory dynamics experimentally observed in the visual system are typically not considered in artificial neural networks (ANNs). Computational models of neocortical dynamics, on the other hand, rarely take inspiration from computer vision. Here, we combine methods from computational neuroscience and machine learning to implement multiplexing in a simple ANN using oscillatory dynamics. We first trained the network to classify individually presented letters. Post-training, we added temporal dynamics to the hidden layer, introducing refraction in the hidden units as well as pulsed inhibition mimicking neuronal alpha oscillations. Without these dynamics, the trained network correctly classified individual letters but produced a mixed output when presented with two letters simultaneously, indicating a bottleneck problem. When introducing refraction and oscillatory inhibition, the output nodes corresponding to the two stimuli activate sequentially, ordered along the phase of the inhibitory oscillations. Our model implements the idea that inhibitory oscillations segregate competing inputs in time. The results of our simulations pave the way for applications in deeper network architectures and more complicated machine learning problems.

## Author summary

Computer vision is a subfield of artificial intelligence focused on developing artificial neural networks (ANNs) that classify and generate images. Neuronal responses to visual features and the anatomical structure of the human visual system have traditionally inspired the development of computer vision models. The visual cortex also produces rhythmic activity that has long been suggested to support visual processes. However, there are only a few examples of ANNs embracing the temporal dynamics of the human brain. Here, we

NIHR Oxford Health Biomedical Research Centre (NIHR203316 to OJ). M.I. acknowledges funding from the grant CAPES/PRINT 88887.583995/2020-00 provided by the Brazilian Government. He is also indebted to the Center for Human Brain Health (CHBH) at the University of Birmingham for their hospitality. MG is funded by the DBI2 project (024.005.022, Gravitation), which is financed by the Dutch Ministry of Education (OCW) via the Dutch Research Council (NWO) as well as a Wellcome Trust Discovery Award (grant number 227420) and the NIHR Oxford Health Biomedical Research Centre (NIHR203316). The funders had no role in study design, data collection and analysis, decision to publish, or preparation of the manuscript.

**Competing interests:** The authors have declared that no competing interests exist.

present a prototype of an ANN with biologically inspired dynamics—a dynamical ANN. We show that the dynamics enable the network to process two inputs simultaneously and read them out as a sequence, a task it has not been explicitly trained on. A crucial component of generating this dynamic output is a rhythm at about 10Hz, akin to the so-called alpha oscillations dominating human visual cortex. The oscillations rhythmically suppress activations in the network and stabilise its dynamics. The presented algorithm paves the way for applications in more complex machine learning problems. Moreover, we present several predictions that can be tested using established neuroscientific approaches. As such, the presented work contributes to both artificial intelligence and neuroscience.

## 1 Introduction

The inclusion of convolution in artificial neural networks (ANNs) was originally inspired by the feature detection properties of cells in early visual cortex, and marked a significant milestone in computer vision [1]. Building on this innovation, deep convolutional neural networks (CNNs) have successfully addressed a wide range of image classification challenges, as demonstrated by Krizhevsky et al [2] in the ImageNet competition (see [3] for review). Moreover, the hierarchically organised representations emerging in these networks have been repeatedly shown to map onto those identified from human MEG and fMRI recordings of the visual ventral stream [4–6] and intracranial recordings from the non-human primate brain [7–11]. Despite the success of embracing the spatial tuning properties of visual neurons for computer vision, there are only few examples of ANNs that have drawn inspiration from the *temporal dynamics* of cortical activity [12–14]. The field of computational neuroscience, on the other hand, has focused on understanding the neural circuitry underlying oscillatory dynamics in cortical and subcortical structures [15–18]. While these biologically informed models have also been linked to cognition [19–21], they have so far not found application in computer vision. For instance, alpha oscillations (8–12Hz) dominate electrophysiological recordings from the human occipital lobe [22, 23], however, their computational benefit has so far not been explicitly explored in ANNs.

Here, we propose a new framework combining principles from computational neuroscience and machine learning to show that dynamic hidden activations allow an ANN to overcome computational bottlenecks when processing simultaneous visual inputs. We deliberately present a tractable network with a reduced architecture to demonstrate the computational benefit of oscillatory dynamics in computer vision. Our aim is to pave the way for applications in deep CNNs that can benefit from both the spatial tuning properties and temporal dynamics of the visual system.

Computational bottlenecks arise in both CNNs and the primate visual system when processing multiple inputs due to their hierarchical, converging architecture [24–29]. Neurons in the early layers of CNNs resemble simple cells in the primary visual cortex that possess small receptive fields, tuned to detect edges and contours in the visual input [1, 30]. The receptive fields of neurons in later layers of the CNN encompass a broader area within the visual field, akin to the inferior temporal (IT) cortex which has been ascribed a critical role in core object recognition [31–34]. While early visual cortex has been shown to process visual features in parallel [35–37], the semantics of multiple visual stimuli have been argued to be extracted serially [38, 39]. Accordingly, the visual and auditory systems in human and non-human primates have been posited to handle simultaneously presented stimuli through "multiplexing" [40], that is, by dynamically switching between the activity patterns associated with each stimulus

[41, 42] (also see [43, 44] for review). Through this mechanism, a single neuron can contribute to the neural code of multiple stimuli [45]. This dynamic interleaving of neural responses to simultaneous stimuli requires precise temporal coordination.

One example of such a time-dependent multiplexed coding scheme is phase coding, which underlies a modulation of spiking activity by ongoing low-frequency neuronal oscillations [14, 46–48] (also see [43, 44] for review). For example, place representations, encoded in spatially distributed firing patterns, have been shown to be ordered along the phase of hippocampal theta (4–8Hz) oscillations [46, 47, 49–51].

Visual perception has been proposed to be supported by a similar mechanism based on neuronal alpha oscillations, which have long been linked to functional inhibition [52, 53]. As the strength of the inhibition waxes and wanes along the alpha cycle, strong inhibition at the peak of the oscillation reduces the probability of neuronal firing in the population [54, 55]. Consequently, only neurons receiving sufficiently strong excitatory inputs fire at the early phases of the alpha cycle [56–58]. As the inhibition reduces toward the trough of the alpha cycle, the neurons may fire successively according to their excitability [56–58]. In this way, the interplay between inhibition by alpha oscillations and neuronal excitability generates a temporal code.

To visualise this process, imagine the following scenario at the supermarket: While searching for the ingredients for a passionfruit martini, your gaze lands on the key element: a passionfruit, placed next to an apple (Fig 1a). With your gaze fixating on both pieces of fruit, your visual system needs to find a way to represent and process them as coherent but separate objects. It is well-documented that object-based attention is associated with increased neuronal excitability [60–65]. As a result, all neurons responding to the features of the passionfruit

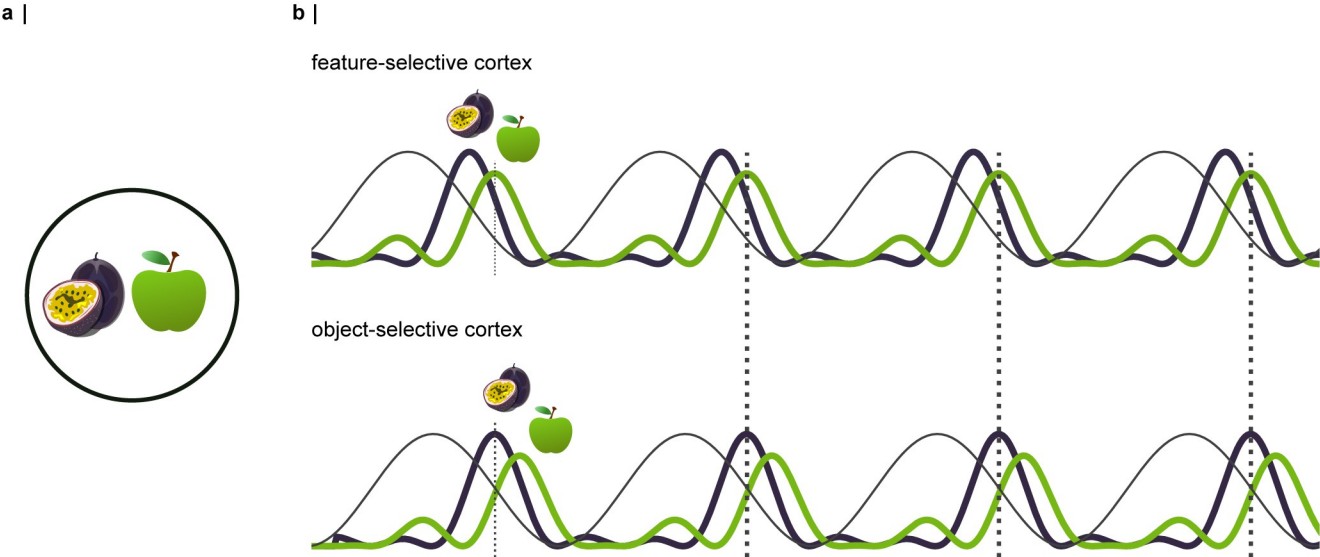

**Fig 1. Concept: An interplay between object-based attention and neuronal alpha oscillations implements a pipelining mechanism reflected by a temporal code [58, 59]. a |** Example: A passionfruit and an apple are competing for the processing resources of the visual system (open-source fruit icons adjusted from openclipart.org). **b |** Inhibitory alpha oscillations modulate neuronal firing rhythmically. The alpha oscillations are illustrated by the black line, the aggregate neuronal responses to the passionfruit and the apple are shown as the deep purple and green lines, respectively. As the neurons responding to the passionfruit will receive stronger excitatory inputs, they will activate at an earlier phase of the alpha cycle compared to the neurons representing the apple. Refraction causes a momentary inactivation of the neural representation, allowing the neurons encoding the features of the apple to activate. As the apple is processed in feature-selective cortex (e.g. V4), its representation has already been passed on to the next stage of the hierarchy, the object-selective cortex (e.g. IT).

receive strong excitatory inputs and will thus overcome inhibition by the alpha oscillations at an earlier phase than the less strongly excited neurons representing the apple (Fig 1b, top panel). It is known that neuronal excitation triggers inhibition [66], for instance, due to the activation of GABAergic inhibition [67] or membrane properties such as the calcium-activated potassium current [68]. Therefore, following the burst of activation associated with the passionfruit, refraction will momentarily deactivate its neural representation. As the inhibition decreases further, the neurons attuned to the apple will take over. As Fig 1b illustrates, these dynamics implement a temporal code, whereby the signals corresponding to the competing items activate successively along the alpha cycle. As the passionfruit is processed earlier in the alpha cycle, its neural representation will reach the next layer of the hierarchy with a temporal advantage over the apple. As the apple is processed in feature-selective cortex, e.g. V4, the passionfruit is represented in the object-selective cortex, e.g. IT cortex. In this way, the visual system might solve bottleneck problems through pipelining: multiple stimuli are processed in parallel, while their representations are segmented in time [58].

In the following, we will present how a mechanism inspired by these concepts can be integrated into an ANN, to allow a successive read-out of competing inputs. We will refer to our model as a dynamical artificial neural network (dynamical ANN). This work serves as a proof of principle to demonstrate the basic principles and analyse the ensuing dynamics in the network. Our model relates to previous work investigating emergent dynamical and oscillatory properties in systems for information storage [69] and Recurrent Neural Networks (RNNs) for image classification [12] and sequence learning [14]. In comparison to these works, we tune the dynamics of the system such that the network is able to segment the representations of competing stimuli along the phase of ongoing oscillations in the hidden units.

## 2 Methods

To implement a dynamical ANN we first trained a two-layer network on a simple image classification task. After training, we added biologically inspired dynamics to the hidden layers motivated by alpha oscillations in the human visual system. The dynamics were not included in the training process and did not change the weights of the network.

### 2.1 Network architecture

We consider a fully connected ANN with two hidden layers, consisting of 64 and 32 hidden nodes, respectively, and an output layer with three nodes (Fig 2a). The inputs to the network were images of size $56 \times 56$, each showing one of three letters ("A", "E", and "T"), presented in one of the image's quadrants (Fig 2b). We aimed to show that integrating oscillatory dynamics into the hidden layers would allow the network to overcome computational bottlenecks when processing an image presenting two letters at the same time. Therefore, we implemented competition between the quadrants by applying a weight matrix of size $28 \times 28$ to the input with a stride of 28. The results of the convolution between each quadrant and the weight matrix were then summed in each hidden node. To make the network dynamics tractable, we refrained from using a conventional CNN architecture, and instead used the convolutional kernel merely to implement weight sharing between the quadrants (but see Discussion). The activation $h_j$ in each unit $j$ of a hidden layer was calculated as:

$$h_j = \sigma(z_j) = \frac{1}{1 + e^{-a(z_j - b)}} \tag{1}$$

with $z_j$ the input to each hidden unit. The input $z_j$ arises from the activation in the previous layer according to $z_j = \sum_i w_{ji} h_i$, with $w_{ji}$ being the weight matrix connecting nodes $i$ and $j$, and

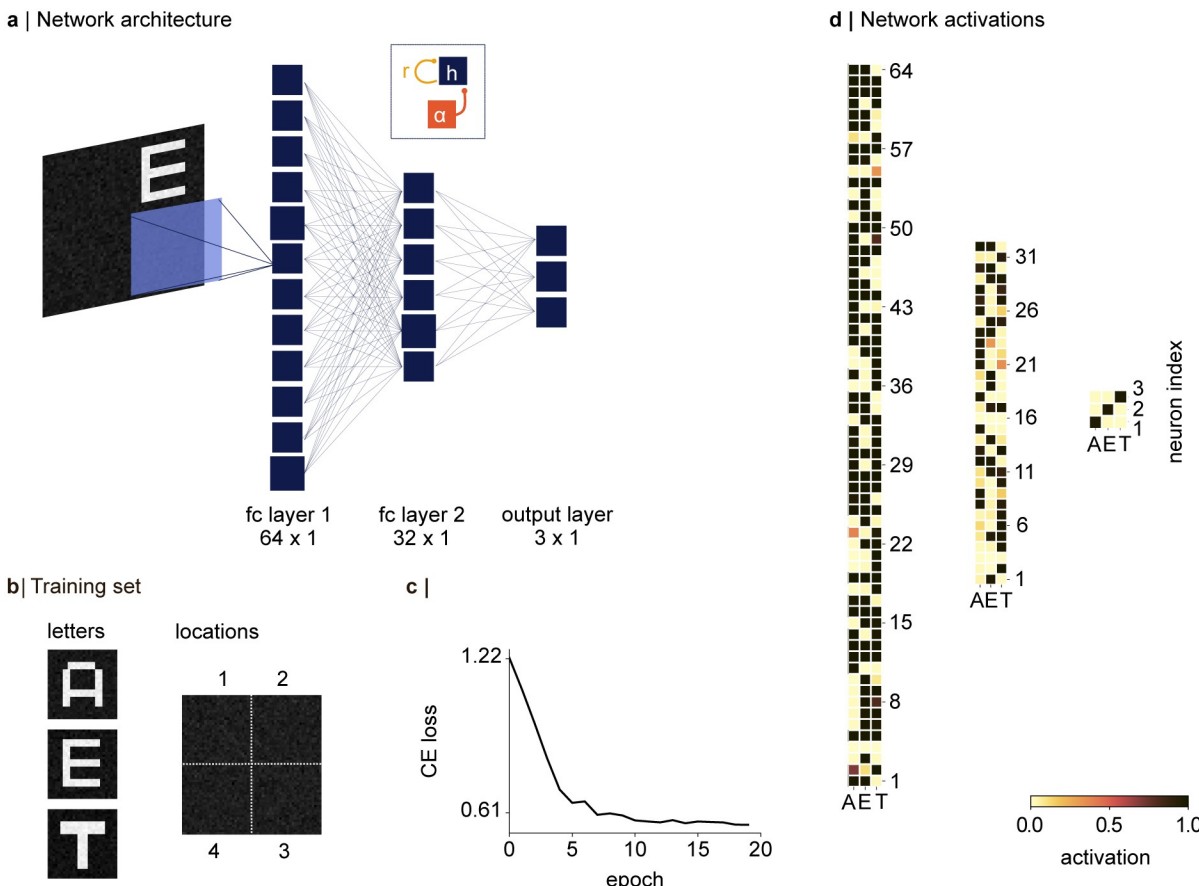

**Fig 2. The classification problem and network architecture. a |** A network with two fully connected hidden layers was trained to classify three letters presented on a $56 \times 56$ image, in one of four quadrants ($28 \times 28$). Convergence of the quadrants in the first hidden layer was implemented by sliding a $28 \times 28$ weight matrix over the image, with a stride of 28. After the training, we added a refraction ($r$ in Eq 4), and pulses of inhibitory oscillations $\alpha(t)$ to each hidden node $h$, as shown in the inset, and Eqs 3 and 4. **b |** The network was trained on a set of 132 images, each showing one of three letters in one of four quadrants. All letters were presented in all quadrants during training. Noise was added to each image as described in the main text. **c |** The network learned to classify the three letters within 20 epochs, approaching a cross-entropy loss of $L_{CE} = 0.567$. **d |** Activations in the hidden layers and the output node in response to three inputs, presented in the three columns. The shifted sigmoid (see main text) resulted in approximately binary activations in the hidden layers. The activations in the output node demonstrate that the inputs are classified correctly.

$h_i$ being either the pixels in the image (for hidden layer 1) or the activations in the first hidden layer (for hidden layer 2). The slope of the sigmoid was set to $a = 2$, and the sigmoid was shifted by the bias term $b = 2.5$. These parameters were fixed, such that a small input ($z \approx 0$) would result in an activation $h$ close to 0, while strong inputs well above 2.5, would result in an $h$ close to 1. Consequently, the activations in the hidden layers were approximately binary ("all or none", see Fig 2d). The activation in each output node was calculated using the softmax function

$$o_j = \text{softmax}(z_j) = \frac{e^{z_j}}{\sum_{j=1}^{K} e^{z_j}} \tag{2}$$

converting the inputs $z_j$ into $K$ probabilities in the output layer [70].

## 2.2 Network training

The training set consisted of three letters, presented in one of four quadrants in the image (Fig 2b). After adding gaussian-distributed noise ranging from 0.01 to 0.25, each input was normalised by its maximum value, such that the luminance in each image ranged between 0 and 1 Fig 2b). The weights of the network were initialised according to a uniform distribution within the range $[-x, x]$, where $x = \sqrt{\frac{6}{n_{\text{in}} + n_{\text{out}}}}$ with $n_{\text{in}}$ and $n_{\text{out}}$ being the number of inputs and outputs to the current layer, respectively (Glorot initialization) [71]. The Adam optimiser was chosen to minimise the cross-entropy loss using stochastic gradient descent [72]. In each epoch, the network was trained on 132 images, whereby each letter appeared at each location on the image. The network weights were learned by backpropagating the error through the network layers (as mentioned above, the bias term was fixed at $b = -2.5$). All experiments reported in Results were conducted on a test set of noisy images with letters "A", "E", and "T" the network had not seen during training.

## 2.3 Network dynamics in the hidden layers

We aimed to implement multiplexing by oscillatory dynamics into a fully connected ANN. To this end, after training the network on an image classification task, we added biologically inspired non-spiking dynamics to each node in the hidden layer, expressed by non-linear ordinary differential equations (ODEs). The ODEs were solved using the forward Euler method with a fixed time step of $\Delta t = 0.001$s. The rate of change in each hidden unit $j$ in the first and second hidden layer was defined as:

$$\tau_h \frac{dh_j}{dt} = -h_j + \sigma\left( \frac{z_j - r_j - \alpha(t) + h_j}{s} \right) \tag{3}$$

where $\tau_h$ determines the timescale at which $h_j$ approaches the sigmoid activation (see Eq 1).

The term $r_j$ approaches $h_j$ with $c \cdot h_j$ with a time delay of $\tau_r = 0.1$s according to:

$$\tau_r \frac{dr_j}{dt} = -r_j + c \cdot h_j. \tag{4}$$

Rhythmic inhibition $\alpha(t)$, mimicking inhibitory neuronal alpha oscillations in the visual system [57], was implemented as a 10Hz sine wave:

$$\alpha(t) = m \cdot [1 + \sin(2\pi t \cdot 10)] \tag{5}$$

with amplitude $m$ being adjustable to modulate the strength and offset of the inhibition. The input $z_j$ to the ODEs in the second hidden layer was calculated from the dynamic outputs of the first layer, and the dynamic activations in the second layer were used to calculate $z_j$ in the softmax activation Eq 2. The results of this feedforward propagation will be explored in Results.

The selection of the parameters was informed by electrophysiological and computational constraints. For instance, the timescale of the activation $\tau_h$ was set to 0.01s, in accordance with the membrane time constant of excitatory neurons of 10–30ms [73–75]. The time constant of $r_j$ was chosen to be $\tau_r = 0.1$s akin to afterhyperpolarization effects caused by calcium-activated potassium currents [76, 77]. The effect of parameters $c$ and $s$ on the dynamics will be explored below (see Fig 3).

## 2.4 Fixed points of the system

The fixed points (steady-state) of $h$ and $r$ are defined by setting Eqs 3 and 4 to zero. Solving for $h$ and $r$ yields:

$$h^* = \frac{z}{c-1} + \frac{s}{c-1}\left[\frac{\log\left(\frac{1}{h^*} - 1\right)}{a} - b\right] \tag{6}$$

$$r^* = c \cdot h^* \tag{7}$$

We found the fixed points numerically, using the function `fsolve` implemented in SciPy [78]. For small values of $s \ll 1$, the equilibrium points could be approximated as $h^* = \frac{z}{c-1}$ and $r^* = \frac{c \cdot z}{c-1}$. Unless otherwise specified, the dynamics of all hidden nodes in the full network were initialised at these approximated fixed points with $z = 2$ (the maximum input size divided by 4, which seemed to be an appropriate initial condition to generate a temporal code).

## 3 Results

This project aimed to show that biologically inspired dynamics will allow a neural network to handle competing inputs, despite having been trained on one stimulus at a time. To illustrate these dynamics, we trained the network to classify three letters "A", "E", and "T" (Fig 2a and 2b). The network learned to read out the images correctly within 20 training epochs (Fig 2c), with the softmax activation in the output layer reaching an activation of 1 in the node of the corresponding image (Fig 2d). Post-training, we integrated the full dynamical system described in Eqs 3 and 4, into the hidden layers. We will first investigate the behaviour of the individual hidden nodes.

### 3.1 Network stability and parameters

Fig 3a demonstrates how the interplay between the input $z = 8.5$ and refraction by $r$ (yellow trace) resulted in dynamical activations in $h$ (blue trace) with a period of approximately 100ms. As $h$ increases, $r$ increases with a delay. As $r$ opposes $h$, the increase in $r$ results in refraction, and the cycle repeats. The adjustable parameters $c$ and $s$ in Eqs 3 and 4 were explored to identify values resulting in robust self-sustained dynamics at a frequency of about 10Hz in the hidden nodes. Fig 3b shows how the frequency of $h$ changes as a function of $c$, for $s = 0.1$, for different levels of $z$ (ranging from 0.5 to 8.5 as indicated by the yellow to black colour scale). The parameter $c$ influences how strongly $r$ grows depending on $h$ (see Eq 4). Consequently, the frequency of $h$ tends to increase as a function of $c$ for $0.5 < z < = 4.5$. There is a slight tendency for the frequency to reduce for $z > 4.5$ (Fig 3c). This is due to $r$ having to oppose a strong input while growing as a function of activation $h$, which is bound at 1 (Eq 1). To induce oscillations in the 10–12Hz range for a broad range of inputs $z$, we selected $c = 10$ (indicated by the dotted box).

Fig 3d illustrates how the amplitude of $h$ decreases as a function of $c$. We aimed to induce dynamics such that $h$ would oscillate between 0 and the sigmoid activation of the current input, which for large values of $z$ is close to 1. This amplitude was achieved for the selected $c = 10$. Note that for $z < 4.5$ the amplitude of $h$ was slightly larger than the sigmoid activation of $z$. For instance, for $z = 1.7$ the activation is $h = \sigma(-2 \cdot (1.7 - 2.5)) \approx 0.17$, however, the amplitude reached values of about 0.8 (see Fig 3d). We did not find this to cause problems when integrating the dynamics into the network, as most activations in the hidden layers were outside the linear part of the sigmoid, and thus approached an activation of 1 (Fig 2c, layer 1; see below for details).

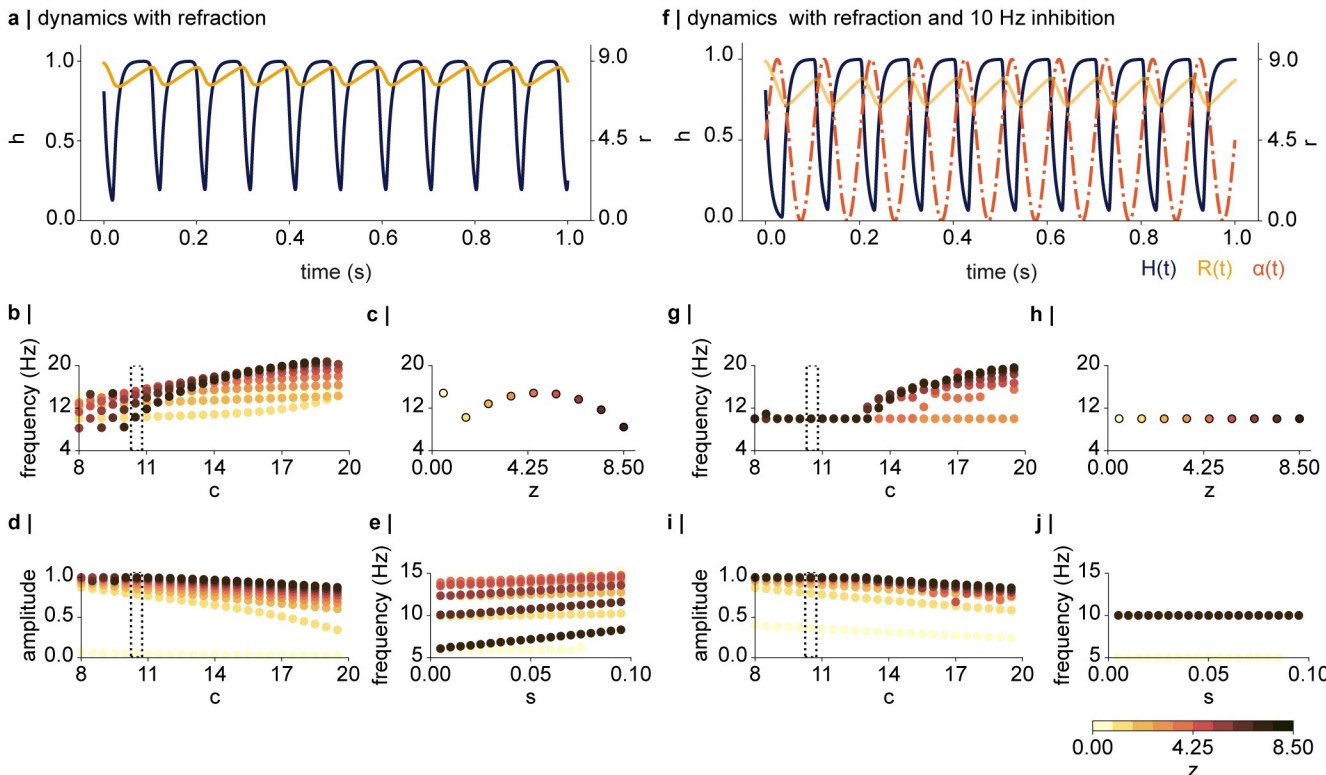

**Fig 3. Exploration of the tunable paramaters *c* and *s* without (a—e) and with (f—j) 10Hz inhibition. a |** Exemplary dynamics in *h* for one input *z* = 8.5, with *c* = 10 and *s* = 0.1 without the 10Hz inhibition. The interplay between *h* (blue trace) and *r* (yellow trace) leads to dynamics with a period of approximately 100ms. **b |** Frequency as a function of *c* for different input values *z* (colour coded from 0.5 o 8.5). High values of *z* require larger values of *c* to generate oscillatory dynamics in *h*. **c |** A value of *c* = 10 (indicated by the dotted box in **b**) leads to dynamics in the 7–13Hz range for *z* > 0.5, whereby the frequency of the dynamics increases approximately monotonically with input size *z*. **d |** Amplitude of *h* as a function of *c*, showing that for high values of *c*, *h* will not reach an activation of 1. Values of *c* < 13 seem appropriate to induce dynamics in the 10–12Hz range that reach the sigmoid activation of 1. **e |** Frequency of *h* as a function of *s*, indicating that an approximately linear increase for all values of *s* within a range of 0.005 to 0.1. **f |** The dynamics of *h* and *r* shown in **a** are entrained by the 10Hz inhibition (dotted orange line). **g-j** The inhibition stabilises the dynamics in the 10Hz range. **g |** For large values of *c* > 13, the dynamics escape the periodic inhibition and oscillate at a faster frequency. **h |** frequency as a function of *z*, for *c* = 10, showing that all nodes with inputs *z* > 0.5 are entrained by the 10Hz rhythm. **i |** The amplitude of *h* as a function of *c* in the presence of the inhibition. The amplitude tends to be slightly larger than the sigmoid activation for small values of *z*. **j |** In presence of inhibition, *h* is robustly entrained to the 10Hz rhythm, for all values of *s* and *z*.

Another instrumental parameter in Eq 3 is *s*, which serves to scale the various input parameters. A small *s* will effectively increase the steepness of the sigmoid, resulting in a more step-like response. Fig 3e depicts the frequency of the node as a function of *s* and suggests that values ranging from 0.005 to 0.1 result in dynamics of about 10Hz for a range of inputs *z*. Based on these observations, we selected *s* = 0.1.

We next explored the effects of adding pulsed inhibition at 10Hz to the hidden dynamics ($\alpha$(t) in Eq 3). As depicted in Fig 3f, the inhibition (orange dashed line) entrained the dynamics, such that *h* activated in anti-phase to the 10Hz rhythm. Another effect of the inhibition is that *r* oscillates between lower values than before. This is sensible, as the inhibition and refraction *r* work together to reduce the activations in *h*. Fig 3g depicts the frequency of *h* as a function of *c*, showing that in the presence of the inhibition, all hidden nodes oscillate at a frequency of 10Hz, when 8 < *c* ≤ 13, for the current input range of *z* ≤ 8.5. For large values of *c* (>13) the dynamics escape the inhibitory rhythm when *z* is sufficiently large (> 5) and oscillate at a faster rate. The frequency in *h* as a function of input size *z* is shown in the right panel, confirming the 10Hz entrainment. Fig 3i shows the amplitude of *h* as a function of

*c*, demonstrating that the amplitude is most stable for $c < 13$. Notably, the amplitude of *h* again reaches values above the sigmoid activation of *z* (as described above), however, we did not find this to interfere with the dynamics in the full network. Lastly, the exact value of *s* within the 0.005 to 0.1 range did not change the frequency, for all $z > 0.5$ (Fig 3j).

We conclude that we could achieve dynamic activations in the hidden nodes in the 8–12Hz range (with and without oscillatory inhibition) for a large range of the parameters. Based on the presented simulations, we settled on the parameters $\tau_h = 0.01$s, $\tau_y = 0.1$s, $c = 10$, $s = 0.1$ in all following simulations, as these produced robust dynamics at approximately 10Hz and an activation *h* close to 1 for a wide range of inputs.

## 3.2 Inhibitory oscillations stabilise the dynamics in the dynamical artificial neural network

After training the network on the "A-E-T" classification problem and exploring the behaviour of the individual nodes shown in Fig 3, we simulated the dynamics in the full network while keeping the weights connecting the layers fixed.

Fig 4a and 4b show the dynamics in the output and hidden layers of the network, in response to a single input letter, A, with $\tau_h = 0.01$s, $\tau_r = 0.1$s, $c = 10$, $s = 0.1$, but without any inhibitory oscillation ($m = 0$, Eq 5). The output node corresponding to A (blue trace) oscillates at approximately 15Hz (Fig 4a). The leftmost panel in Fig 4b depicts the activations in *h* in layer 1 as a function of time, for *z*'s ranging from 0.5 to 8.5 (only unique values of *z* are shown). The dynamics demonstrate that the phase of the oscillations depends on the size of input *z*, leading to inconsistent phase delays between the network nodes. Due to the softmax activation introducing lateral inhibition in the output nodes, these phase delays cause a spurious intermittent activation of the output node corresponding to letters "E" and "T" (orange and green traces). The right panel in Fig 4b, depicting *h* as a function of *r*, shows the limit cycle and fixed points (indicated by the diamond-shaped scatters) for each input *z*. All units demonstrate a limit cycle behaviour but with different amplitudes.

The bottom panel shows the dynamics in the second layer. While the phase-locking between the hidden nodes in the second layer is slightly stronger compared to layer 1, activations in the nodes receiving smaller inputs start to lag the nodes receiving larger inputs after about 300ms. The right-most plot, showing the relationship between *h* and *r*, again suggests a limit cycle behaviour, although the amplitude of each node appears to vary over the course of the simulation. This is likely due to the second layer receiving dynamical, non-phase-locked inputs from the first layer.

Introducing oscillatory entrainment by periodic inhibition into each layer of the network stabilises the dynamics in the entire network. Fig 4c shows the dynamics in the ouput layer, with the amplitude of the oscillatory drive set to $m = 0.5$, and a phase delay of $\Delta\phi = 0$ between the layers. Comparison to the dynamics without the oscillatory drive shows that the inhibition removes the spurious activation in the output nodes corresponding to "E" and "T" (orange and green trace). This stabilisation of the read-out underlies increased synchrony both within and between the hidden layers, as indicated in Fig 4d. In particular, the phase-locking between the activations in the first layer has been notably increased by the oscillatory drive; as shown in the time course of the activations in *h* shown in Fig 4d, top left. Comparison of the limit cycles shown in Fig 4b top right, and Fig 4d top right, demonstrates a wider limit cycle in presence of the oscillatory inhibition, reflecting an increased amplitude of the dynamics in *h*.

The dynamics in the second layer also show higher synchrony and stabilised trajectories *h* and *r* in presence of the inhibitory oscillations (Fig 4d, bottom right). This underlies the inhibitory oscillations applied to the second layer and the more synchronous inputs from the first

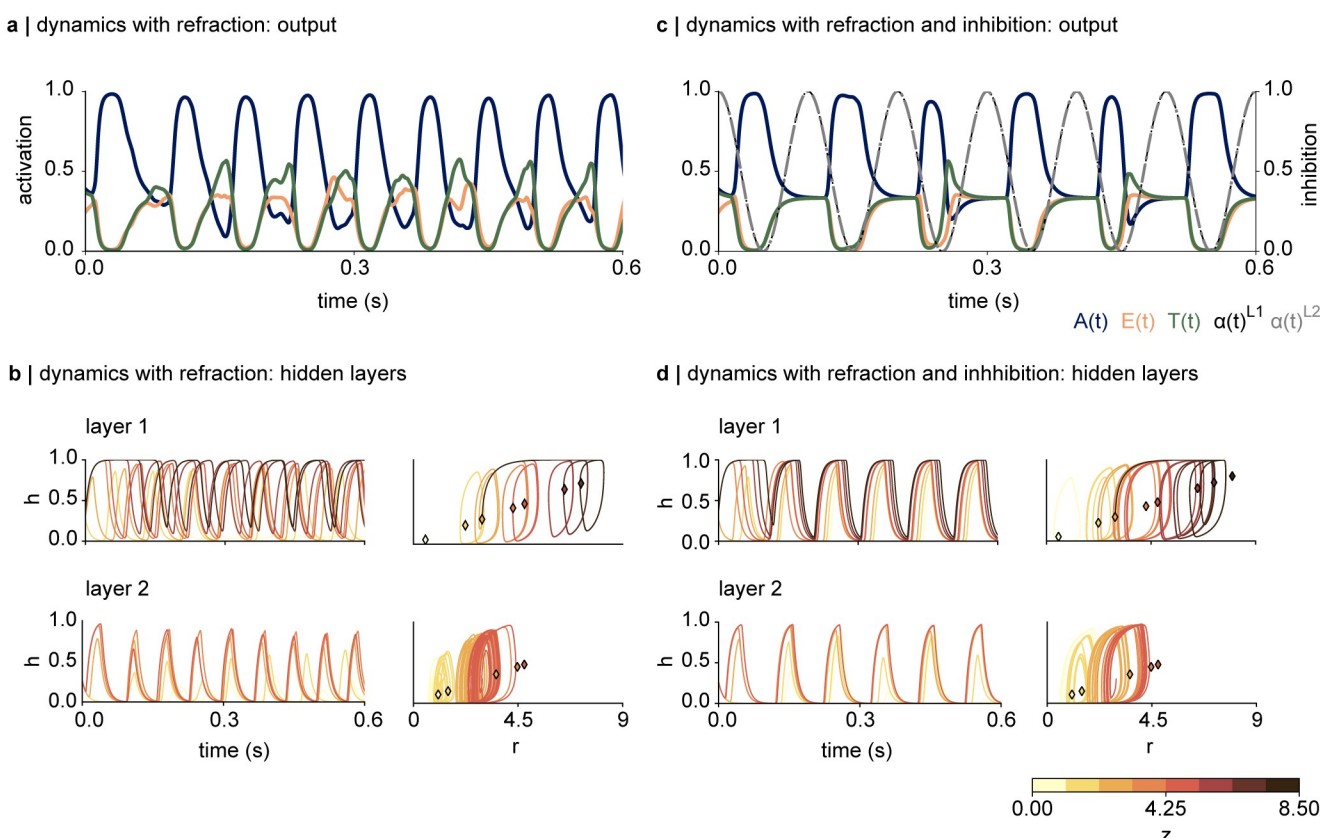

**Fig 4. a-b** Dynamics underlying the interplay between $h$ and refraction $r$ (Eqs 3 and 4), without the inhibition. **a |** The output node corresponding to the letter A oscillates at about 12Hz. The system spuriously activates the output node E and T. **b |** (top left) The activations in the first hidden layer show that the frequency of the oscillation depends on the input size $z$, as suggested in Fig 3c. (top right) Trajectory of $h$ as a function of $r$, with the fixed points indicated by the coloured diamonds. The dynamics are attracted towards a stable limit cycle, depending on the magnitudes of the input, and orbit around the fixed point. (bottom left) Activation of $h$ in layer 2 as a function of time. There is a notable phase lag between the network nodes based on the input size $z$. (bottom right) The amplitude of the activation in each node appears to decrease over time, as indicated by the inward spiralling of the limit cycle. **c-d |** Periodic inhibition stabilises the dynamics in the system. **c |** Periodic inhibition at 10Hz with an amplitude of $m = 0.5$ was added to each layer. The read-out of the presented letter oscillates in anti-phase to the inhibition. **d |** 10Hz inhibitory oscillations notably stabilise the dynamics within and between the network layers, as indicated by the phase-locking between the nodes, and the stabilised H-R trajectories for the activations in layer 2.

layer. Comparison of the plots at the bottom right in Fig 4b and 4d reveals that the amplitude of the activation in the second layer varies less in the presence of the inhibition. These simulations show that the oscillatory inhibition stablises the dynamics of the multi-layer network.

## 3.3 Simultaneous presentation of two inputs produces a temporal code

**3.3.1 The bottleneck problem in absence of the dynamics.** The network correctly classified individually presented stimuli after the training, as demonstrated in Fig 2c. However, when presented with two stimuli simultaneously (Fig 5a), the network produced a mixed output. The right panel in Fig 5a shows the network activations in response to all possible combinations of stimuli. Fig 5b shows the corresponding time course to these simultaneous inputs, which was achieved using Eq 3, with $s = 1$, $r = 0$, $\alpha(t) = 0$, with the network dynamics initialised at $h = r = 0$ in all hidden nodes. The network distributes the activations in the output layer over the nodes corresponding to the respective inputs, suggesting that the output layer produces a weighted average of the activations to both stimuli. The network further responds

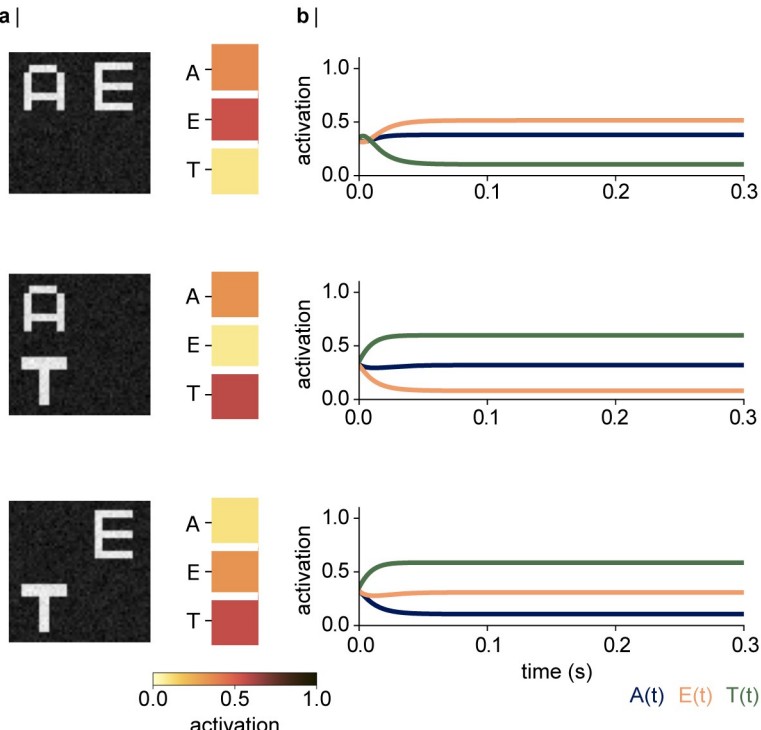

**Fig 5. The bottleneck problem when the network receives two inputs simultaneously. a |** (left column) Exemplary input combinations of the two letters. (right) Activations in the output layer are distributed over the respective nodes. The network further appears to show a preference for letter "E" over "A", and "T" over both "A" and "E", despite the letters being presented at the same luminance. **b |** Time course of the activation in the output layer, for $s = 1$, $r = 0$, and $\alpha(t) = 0$. The shared output is a consequence of the two items competing.

more strongly to the letters "T" and "E" compared to letter "A". This indicates a bottleneck problem when the network needs to classify two stimuli, as well as an inherent preference for "T" over "E" and "A". In contrast, the abilities of our visual systems to recognise a stimulus do not decrease with the number of objects. As such, the visual system must be able to segment the representations of the different stimuli. As outlined above, it has been proposed that multiplexing in the visual system may underly oscillatory dynamics. In the following, we will explore how the complete dynamical network presented in Fig 4 and described by Eqs 3 and 4 responds to simultaneous stimuli.

**3.3.2 Refraction and 10Hz inhibition allow read-out of competing stimuli as a temporal code.** Fig 6b shows how the output layer of the dynamical network responds to two simultaneous inputs; "A" and "E" (top), "E" and "T" (middle), and "A" and "T" (bottom). To mimic spatial attention to letter "A", we multiplied the pixels of an image showing letter "A" (without noise) with 1.2, and the pixels defining the letter "E" by 0.8 (Fig 6a, top panel). After adding noise to the image as described in 2 Methods, we divided the image by its maximum value to scale the luminance between 0 and 1. We repeated the same for the other input combinations (Fig 6a middle and bottom panel, zoom in to see noise). Fig 6b shows the dynamics in the output layer, based on the simulations with parameters $\tau_h = 0.01$s, $\tau_r = 0.1$s, $c = 10$, $s = 0.1$, without any oscillatory drive. The node corresponding to "A" activates first (blue trace), followed by the node corresponding to "E" (orange trace). Subsequently, the network continues to alternate between the two inputs, whereby the nodes corresponding to the presented letters reach

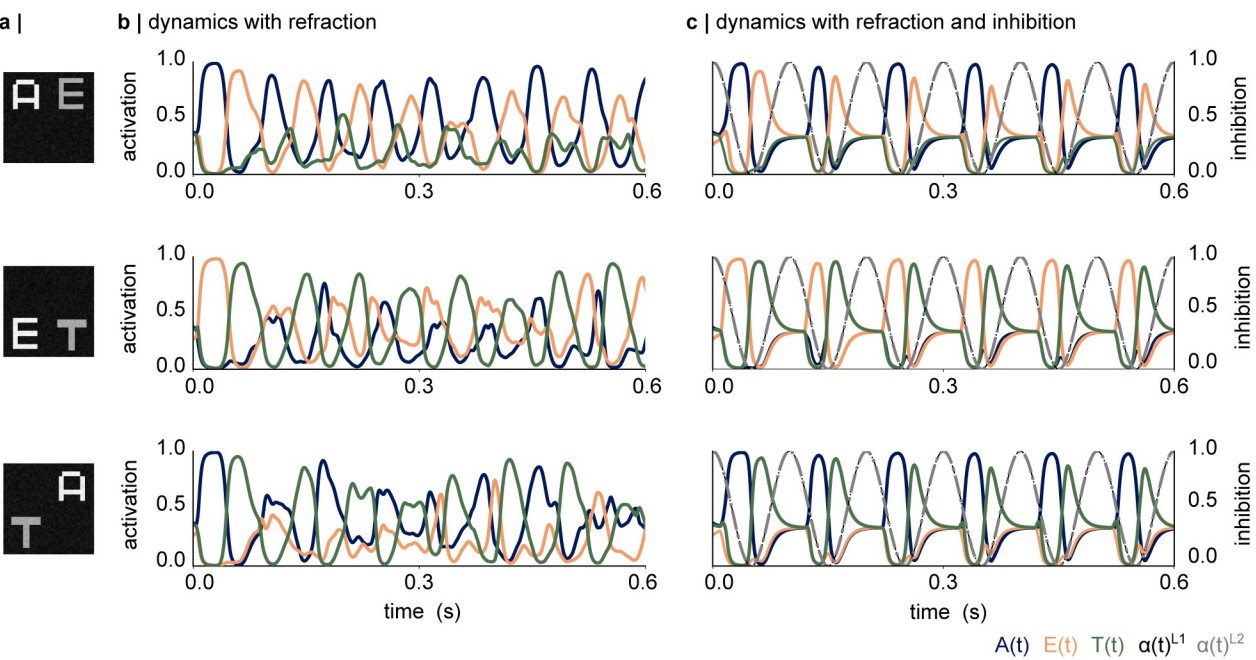

**Fig 6. The dynamical artificial neural network (dynamical ANN) multiplexes simultaneously presented stimuli. a |** Examples of the simultaneous inputs. Attention towards the letter "A" was mimicked by increasing the input of all pixels belonging to "A" while reducing the inputs of the letters "E". This was repeated for all other input combinations. **b |** The interplay between the excitation and refraction results in dynamic activations in the output layer, whereby one of the two letters is read out at a time. However, the dynamics show notable instability, and the node corresponding to the "attended" input E activates over longer periods of time. **c |** Introducing pulses of inhibition (alpha oscillations) into the network generates a temporal code in the output layer, with the letters being read out along the phase of the alpha oscillation according to the input gain.

activation levels between 0.8 and 1. As such, the interplay between activation and refraction implements a multiplexing mechanism. However, the dynamics of the multiplexed representations are unstable, as indicated by variations in the amplitude of the traces, and a spurious activation of the letter "T", which was not presented. The simultaneous presentation of "E" and "T" (Fig 6b, middle panel) and "A" and "T" (Fig 6b, bottom panel) results in similar dynamics with a varying amplitude and spurious activation of the letter that was not presented in the image.

Introducing the oscillatory inhibition results in stabilised dynamics, whereby the simultaneous inputs are read out as a temporal code, organised along the phase of the inhibitory oscillation (Fig 6c). We replicated these dynamics for all combinations of simultaneous inputs (two letters at a time), as shown in S1 Fig. The maximum read-out accuracy (activation) for each input is indicated in each plot. As the activations were calculated using a softmax function, a value of 0.99 (e.g. for letter "A" in top left panel), indicates that the network is 99% certain about the presence of the letter "A" while the remaining 1% are shared between letters "E" and "T". While the response to "E" in the combined "T" and "E" input is notably reduced compared to the other experiments (S1 Fig bottom right), the network still achieves a read out accuracy of 0.59, well above the chance value of 0.33. The simulations show that the network is able to segregate all input combinations in the test set.

We hypothesised that speeding up the refraction would allow an increase of the number of items within the temporal code. This was tested by repeating the simulations shown in Fig 6 with $\tau_r = 0.05$s. However, while a reduced time constant for the refraction did result in a faster

activation of the two nodes corresponding to the letters in the image (S2b Fig), only the first inhibitory cycle showed three activations, whereby the attended letter is read out before and after the unattended one (S2c Fig). A faster refraction was further associated with an overall reduced amplitude, and an occasional activation of the output node corresponding to the letter that was not presented in the image. Overall, increasing the frequency of the refraction did not seem to offer a stable solution for increasing the number of items in the temporal code.

Reducing the frequency of the inhibitory oscillation to 5Hz, leads to a robust temporal code with three activations per cycle of the inhibitory oscillation (S3c Fig). While these simulations briefly activate the letter not presented (S3c Fig top and middle panel), slowing down the inhibition appears more effective in increasing the number of items in the phase code than speeding up the refraction.

Following these tests, we investigated whether the network could generate a temporal code representing all three stimuli. S4a Fig shows the exemplary input, which was generated by multiplying the attended letter ("E") by 1.2, the unattended letter("T") by 0.8, and "A" with 1. After adding the noise, the image was scaled to the luminance range from 0 to 1. We used the original settings for the dynamics with $c = 10$, $s = 0.1$, $\tau_h = 0.01$s, and $\tau_r = 0.1$s. Indeed, refraction without inhibition allows the network to dynamically activate each letter in the input, albeit with varying amplitude and activation period (S4b Fig). Generating a temporal code with all three items, however, proved to be challenging. Introducing a 10Hz inhibition resulted in "E" and "A" being read out in the first and second cycle of the inhibition, respectively, after which the network produced a code with two items ("E" and "A") in each cycle (S4c Fig). Slowing down the inhibition to 6Hz resulted in a temporal code with three items in two out of the five cycles shown here, however, the network often activated the output node corresponding to "E" after reading out "A" (S4b Fig). In sum, while the network was able to produce a stable temporal code with two inputs, it was not trivial to produce a code with three stimuli. We will explore the biological relevance of increasing the number of items in the temporal code in Discussion.

**3.3.3 Dynamical artificial neural network shows both parallel and serial processing.** A more detailed investigation of the multiplexing dynamics demonstrated in Fig 6 is shown in Fig 7 for each layer individually; for the exemplary simultaneous input "E" and "T". The top panels in Fig 7a and 7b indicate how strongly the hidden representations to the combined input correspond to the neural representations of both letters individually. For instance, for letter "E" (orange trace) this measure of similarity was calculated as:

$$s_E(t) = \frac{\mathbf{h}^{ET}(t) \cdot \mathbf{h}^E}{\mathbf{h}^E \cdot \mathbf{h}^E} \tag{8}$$

with $\mathbf{h}^{ET}(t)$ being the activations in the respective hidden layer at time point $t$, to the simultaneously presented letters "T" and "E" and $\mathbf{h}^E$ being the activations in the hidden layer to the letter "E" in the trained, non-dynamical network (see Fig 2c). A similarity value of $s_E(t) = 1$ in Eq 8 indicates that all hidden nodes corresponding to the individual letter "E", are activated by the current image showing both letters simultaneously.

In the first layer, the normalised dot product indicates that the nodes representing both "E" (orange trace) and "T" (green trace) activate in parallel, in anti-phase to the inhibitory oscillation (Fig 7a top panel). The network also appears to activate the hidden representation of "A", albeit to a lesser extent (blue trace). Indeed, the time course of the activations in each node (Fig 7a, bottom panel) demonstrates that almost all nodes in the first layer activate during the excitatory cycle of the oscillation. This indicates that the first layer represents the two presented letters in parallel.

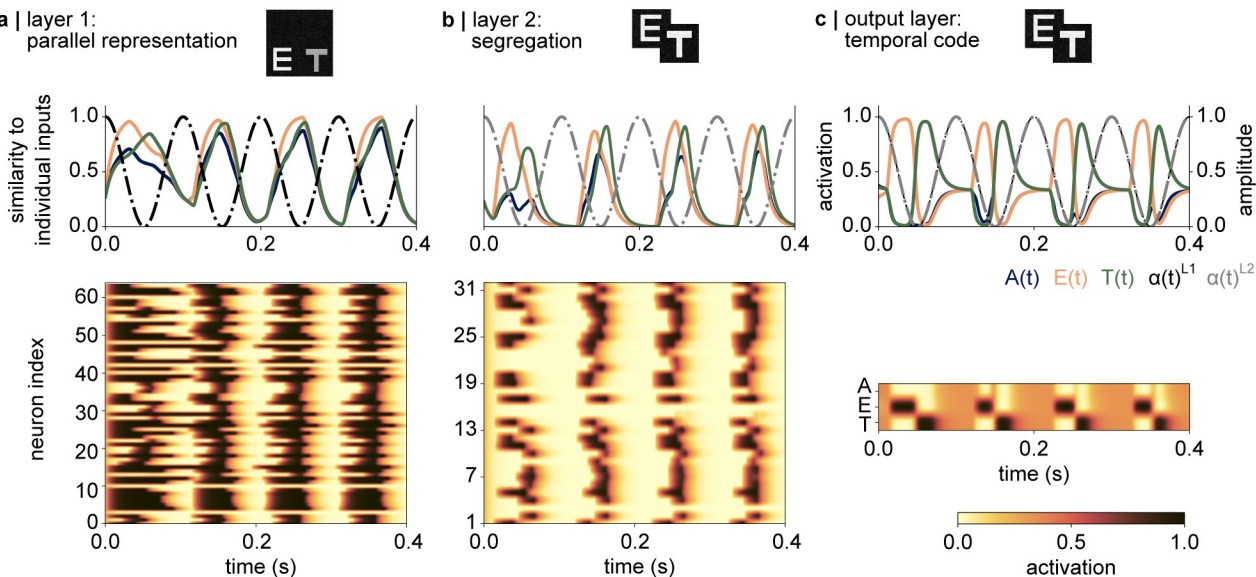

**Fig 7. Parallelisation and segmentation of the competing inputs along the network layers. a |**(top) The time course of the representations of simultaneous inputs "E" and "T" in layer 1. The normalised dot product between the current activation and the activation in response to an individually presented letter "E" (orange trace) and "T" (green trace") indicates that both stimuli are represented in parallel. While the similarity to the attended letter "E" increases slightly faster, both traces reach the maximum value of 1 at about the same time. The activations also show a similarity to letter "A" which is not presented in the image. (bottom) The time course of the activations of all neurons in the layer, indicating that almost all neurons activate at a similar time in anti-phase to the inhibitory oscillation. **b |** Segregation of the simultaneous stimuli can be observed in the second hidden layer. (top) The normalised dot product suggests that the nodes corresponding to each input activate at different phases relative to the 10Hz inhibition. (bottom) The temporal segmentation observed in the top panel can also be observed as a successive activation of the nodes in the network. The 10Hz inhibition silences the activation in the entire layer. **c |** As shown in Fig 6 middle panel, the softmax activation in the output layer reflects a temporal code, whereby the inputs are read out along the phase of the inhibition, ordered according to the magnitude of the input.

In comparison, the activations in the second layer demonstrate that the nodes responding to each letter are activated in a sequence: the normalised dot product between the current representations and the activations to an individual letter "E" precede the ones corresponding to letter "T" (green trace, Fig 7b). The bottom panel in Fig 7b indicates that a smaller fraction of the network is activated at each time point, and the successive activation of the hidden nodes can be observed. Finally, Fig 7c shows the read-out in the output layer, confirming that the representations of "E" and "T" are fully separated during the excitatory cycle of the inhibition (also see Fig 6c).

In sum, our simulations show how dynamics driven by excitation and refraction enable a fully connected neural network to multiplex simultaneous inputs—a task it has not been trained on explicitly. This mechanism is further stabilised by pulses of inhibition at 10Hz, akin to alpha oscillations in the human visual system.

## 3.4 The phase delay of the inhibition between the layers only has a marginal effect on the temporal code

Previous research on neural codes in the visual and auditory system has suggested that the phase of low-frequency oscillations in electrophysiological recordings carries information about the sensory input [79–81]. The communication through coherence (CTC) theory predicts that the phase relationship between two populations is critical for their communication [82, 83] (also see [84, 85] for computational implementations). While CTC was initially

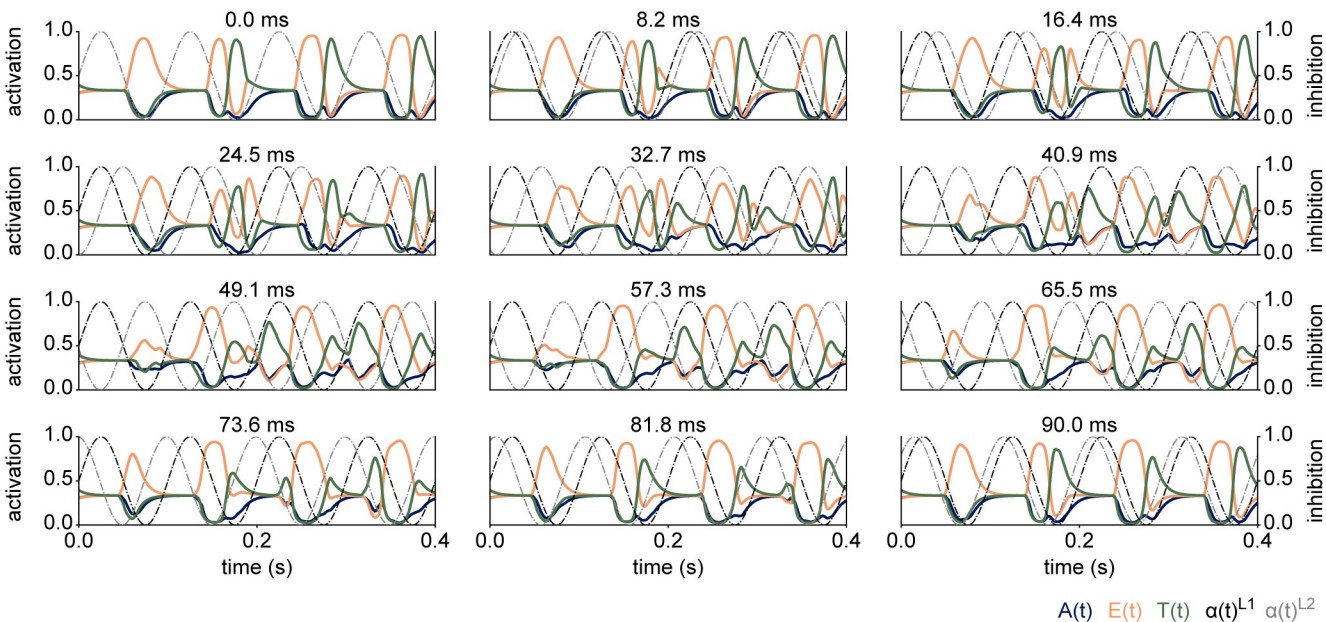

**Fig 8. The temporal code as a function of phase delay between the layers, indicated in the title (in ms).** The temporal code is robust to changes in phase in layer 2, suggesting that the read out depends more strongly on layer 1.

proposed for oscillations in the gamma-band [82, 83], related ideas have been explored for the alpha-band [86]. Based on these concepts, we next tested how different phase delays between the network layers impact the temporal code. Fig 8 shows the temporal codes for different phase delays between the inhibitory oscillations in layer 1 and layer 2. Visual inspection suggests that only a large phase delay of 57.3 and 65.5ms (second to last row) affects the read out of the second stimulus. One might have expected an anti-phase delay between layers 1 and 2 to cut off all communication, however, the network is still able to identify both stimuli. Based on this experiment, we reasoned that the phase of the inhibition in the first layer may affect the temporal code more strongly, as the dynamics in layer 1 are propagated to layer 2 and the output layer. Indeed, shifting the phase of the inhibition in the first layer appeared to affect the read-out of the stimuli in the first two cycles more dramatically (S5 Fig). Still, the temporal code seemed to recover within three cycles of the simulation.

We conclude that the phase of the inhibition may mainly affect the initial conditions of the Euler integration for the selected parameters, but did not interrupt the communication between the layers.

## 4 Discussion

We here demonstrate how integrating semi-realistic neuronal dynamics into an ANN enables multiplexing during image classification. The network was first trained to classify individual letters presented in quadrants. Post-training, the network correctly recognised the individual inputs. When presented with two letters simultaneously, however, the output layer produced a mixed representation of both stimuli due to the bottleneck in the network. Adding refraction to the nodes of the network resulted in alternating activations to the simultaneously presented inputs, suggesting that the network was able to segregate the representations. However, the dynamics expressed notable instability and asynchrony within and between layers. Adding oscillatory inhibition to the layers, akin to alpha oscillations in the visual system, stabilised the

dynamics in the hidden layers. When two inputs were presented simultaneously, the interplay between activation, refraction, and pulses of inhibition resulted in a stable multiplexed code, whereby the output nodes were activated sequentially based on the strength of the input.

Our simulations provide an implementation of the idea that inhibitory alpha oscillations in visual cortex serve to support the processing of simultaneously presented stimuli, by segmenting them into a temporal code [57, 58]. As the inhibition is strongest at the peak of the oscillations, only neurons receiving sufficiently strong excitatory inputs will be able to activate [54–56]. In turn, the neural representations associated with the attended stimulus will overcome the inhibition at an earlier phase of the alpha cycle than the unattended object. This allows temporal segmentation and thus multiplexing of simultaneously presented stimuli [57]. A key contribution of our approach is that we translate conceptual ideas based on neuroscientific studies into a computational model.

Our network embraces two key properties of visual perception: parallelisation and segregation. Prior work has shown that simple visual features can largely be processed in parallel [35–37], while object recognition has been demonstrated to be supported by serial processes [39, 87]. This indicates a bottleneck problem which has been argued to arise from the converging hierarchical structure of the visual system [29, 37, 39, 58]. A similar bottleneck problem arises in our non-dynamic network (Fig 5). By segmenting individual object representations in time, alpha oscillations have been suggested to allow the attended stimulus to pass through the network layers with a temporal advantage over the unattended stimulus (Fig 1, also see [58]). These parallel and serial properties resulting from multiplexing can be observed in our network simulation as well. In the first layer, the stimuli are represented in parallel, while the second layer begins to segment them along the phase of the oscillation (Fig 7). The simulations presented here show how integrating biologically inspired dynamics into a neural network results in a system that can multiplex simultaneous stimuli while embracing the parallel and serial properties of visual object recognition.

Our algorithm is inspired by evidence for phase-coding observed in recordings from the rodent and human hippocampus, whereby spiking activity has been shown to be modulated along the phase of ongoing theta oscillations (4–8Hz [46, 47, 51, 88, 89]). The order in which a sequence of inputs has been experienced, has further been proposed to be preserved in the spiking activity [46, 51] but see [14]. We here demonstrate how the visual system may utilise a similar mechanism based on inhibitory alpha oscillations to support object recognition.

As outlined in Section 2, we tuned the hyperparameters in our simulations to resemble oscillatory dynamics observed in electrophysiological recordings. The rise time of the activation $\tau_h$ was chosen based on the membrane time constant of excitatory neurons (10—30ms) [73]. The activation period of an individual letter within a temporal code was 23–30ms (Fig 6b and 6c), i.e. 35 to 40Hz. This corresponds to the period of gamma oscillations, which have been proposed to be involved in the feedforward processing of visual information [90–92]. As such, our algorithm is strongly linked to the idea that visual processing is modulated by an interplay of gamma and alpha oscillations [57, 92].

A rich body of literature backs the involvement of these oscillations in organising visual processing. For instance, intracranial recordings from the visual cortex in non-human primates have revealed that the phase of spontaneous alpha oscillations modulates spiking activity [54, 93–95] and neuronal gamma oscillations [90, 91, 96, 97]. The phase-amplitude coupling between alpha and gamma oscillations during visual processing has further been replicated using intracranial and MEG recordings from the human brain [98, 99]. Recent human EEG recordings have additionally posited that alpha waves travelling from occipital to frontal areas are actively involved in visual processing [100, 101] (but see [102] for a critical perspective). Similarly, ECoG recordings from marmoset visual cortex have linked travelling waves in the

dorsal and ventral stream to visual performance [103]. Travelling waves have also been shown to modulate neural processing along the visual hierarchy following saccade initiation [104]. These reports show that alpha oscillations may propagate over cortical areas involved in visual processing to coordinate neural activity. However, it has not yet been determined whether this modulation results in successive activations of the neural representations that are in line with our model simulations.

In our model, the number of items in the temporal code could be increased by reducing the frequency of the inhibitory oscillation (S3 Fig). This simulation suggests that the visual system may prolong the period of the alpha oscillations in anticipation of complex visual inputs. So far, however, only an increase in alpha frequency has been linked to visual detection and processing speed in temporal attention paradigms [105, 106]. Since alpha oscillations have been proposed to support saccadic previewing through multiplexing [58], increasing the number of items beyond two may not be necessary or beneficial. According to this model, the first item in the temporal code represents the fixated stimulus, while the second item may represent the goal of the next saccade [58]. Therefore, while changing the dynamics may be relevant for computational goals in the dynamical ANN, we believe that the temporal code with two items organised by inhibitory 10Hz oscillations may capture the dynamics of visual cortex and associated conceptual models more accurately.

Based on the presented model, we propose two testable predictions: First, neural representations may activate along the phase of spontaneous alpha oscillations, ordered according to attention or salience [56, 57]. This prediction can be tested using electrophysiological recordings during visual tasks with multiple stimuli. The neural representations of each stimulus could be extracted from these data using decoding methods such as multivariate pattern analysis [107] or linear discriminant analysis [108]. For instance, using MEG, van Es et al. [109] have recently investigated the effect of ongoing alpha oscillations on the decoding accuracy of visual stimuli in a spatial attention task. The authors have demonstrated that the phase of alpha oscillations in the frontal eye field and parietal cortex of the human brain modulated the decoder's performance. However, a phase delay between the attended and unattended stimuli has not explicitly been reported. Alternatively, the idea that different visual stimuli are represented at different phases of ongoing alpha oscillations could be tested based on intracranial recordings from the mouse brain. Using Neuropixels probes, spiking activity and local field potentials can be recorded intracranially from several cortical areas in mice (and non-human primates, [110, 111]. The mouse visual system has been shown to exhibit a hierarchical structure similar to the one observed in primates [112, 113]. As such, these data could be used to test whether spiking activity is segmented along the phase of ongoing alpha oscillations, for instance, to distinguish a figure from a background [114].

The second prediction of our model is that a stable temporal code emerges from approximately synchronous oscillations in the consecutive network layers. Recent work using concurrent iEEG and MEG recordings has suggested that interactions between alpha oscillations in prefrontal cortex and mediodorsal thalamus mediate visual performance [115]. In light of the literature on travelling alpha waves [100, 101, 103, 104, 116–118] this begs the question of whether neuronal processing along the visual hierarchy is controlled by one driving force as suggested by our simulations, e.g. the thalamus or prefrontal cortex, or a travelling wave propagating forward or backward along the visual hierarchy. With recent advances in brain-wide recordings in mice using Neuropixels [110], it may be possible to investigate whether the driving force of these travelling waves can be established.

Our network relates to previous computational models that have explored the role of biologically plausible dynamics for multiplexing and inter-areal communication [40, 85, 119–121]. We expand on this work by demonstrating how multiplexing and communication

through synchronous oscillatory activity can enhance the computational versatility of a neural network in the context of multi-item image classification. As we aimed to provide a proof-of-principle, we trained the network on a comparably simple classification problem. While the simple nature of the network limits its computational abilities, it allowed a tractable implementation and comprehensive exploration of the imposed dynamics, as demonstrated in Figs 3, 4, 6 and 7. As such, the presented work sets the stage for applying the presented principles to CNNs with a deeper architecture that can solve benchmark image classification problems such as (E)MNIST [122, 123], CIFAR-10 [2], and ImageNet [124]. One technical detail to consider is that modern DNNs typically implement non-linearities using the rectified linear unit (ReLU) function which reduces the vanishing gradient problem in deep architectures and speeds up learning [70, 125]. Since ReLUs do not bind the activations between 0 and 1, a different set of ODEs will be needed to describe the dynamics in future versions of this model.

Alternatively, the dynamics could be integrated in additional layers with sigmoid-like activation functions, between the trained network layers, as conventionally done in spiking neural networks [126, 127]. For instance, Sörensen et al [126] integrated spiking dynamics into a pre-trained CNN, which allowed the network to find a target stimulus in a complex natural image, an ability the model did not exhibit without the spiking dynamics. As alpha oscillations have been shown to modulate spiking activity [54, 93, 94], it would be interesting to understand to what extent oscillatory dynamics could serve to modulate activations in spiking neural networks. By incorporating spiking or non-spiking dynamics into extra layers with activations constrained between 0 and 1, the concepts presented here could be explored in pre-existing deep neural networks. In sum, future versions of this network will expand to deeper architectures and modern image classification benchmarks.

The rate of change in the network nodes was defined by a set of ODEs, following conventional practice in computational neuroscience [128]. ODEs have also found applications in the development of RNNs. For instance, in the form of neural ODEs [129], liquid-time constant neural networks [130, 131], and RNNs [12]. These networks had great success in learning long-range dependencies in time series data [129–131], sequences of images [14], and image classification when the pixels of the input are transformed into a time series (sequential MNIST) [12, 130]. Moreover, Liebe et al. [14] have demonstrated emerging oscillatory dynamics when training an RNN to memorise a sequence. A key difference to our work is that the inputs and outputs in these previous studies were dynamic, which may have resulted in emerging dynamics with minimal intervention by the researcher. However, it would still be interesting to test whether training a network to not only classify images but also to convert simultaneous stationary inputs into a sequence results in rhythmic activations in the hidden layers. It should be emphasised however, that we here provide an implementation of the idea that top-down control by inhibitory alpha oscillations supports multiplexing. This oscillatory top-down control has been proposed to reach the sensory systems of the human and primate brain through thalamo-cortical connections [15] or as a backwards travelling wave initiated in frontal regions [132]. The logic of imposing the dynamics after the training was based on the notion that learning to recognise different objects, i.e. the main task of the visual ventral stream, is different to learning to represent items in a sequence. As such, we argue that the current implementation with imposed external top-down control to learned representations of visual objects is more biologically realistic for the presented problem. In sum, for future extensions of the presented algorithm, it would be interesting to explore if and how biologically plausible dynamics could be used to support the training process.

## 5 Conclusion

We here present a proof-of-concept showing that integrating oscillatory dynamics based on excitation, refraction, and pulses of inhibition into the hidden nodes of an ANN enables multiplexing of competing stimuli, even though the network was only trained to classify individual inputs. Our simulations predict that the visual system of humans and non-human primates handles the processing of multiple stimuli by organising their neural representations along the phase of inhibitory alpha oscillations. These predictions can be experimentally tested using simple attention paradigms and electrophysiological recordings in humans, non-human primates, and rodents. Future versions of the network will include extensions to deeper architectures and modern image classification benchmarks.

## Supporting information

**S1 Fig. Examples of the temporal code for all input combinations.** The first letter in the title is the one for which input gain has been increased. The coloured text indicates the read-out accuracy for each of the presented letters.
(TIF)

**S2 Fig. The dynamics with accelerated refractory dynamics. a |** The experiment was performed on the same input combinations shown in Fig 6. **b |** Speeding up the refraction results in faster activations of the nodes corresponding to the two letters presented in the image. The dynamics again show some instability with a fluctuating amplitude and an occasional spurious activation of the letter that is not presented in the image. **c |** In presence of 10Hz inhibition, the attended letter is read out twice within one inhibitory cycle: once before, and once after the the unattended letter. Following that, the output again shows a temporal code with two items per cycle.
(TIF)

**S3 Fig. The dynamics with longer periods of inhibition. a |** The experiment was conducted on the same input combinations shown in Fig 6 and S2 Fig. **b |** Dynamics with $\tau_r = 0.1$s (same as Fig 6b). **c |** Slowing down the inhibition to 5Hz results in a temporal code with three items. Interestingly, in the second inhibitory cycle, the unattended letter seems to be read out first along the phase of the inhibition. This appears to be due to the brief, low-amplitude activation of the node corresponding to the attended item during the peak of the inhibition. The code stabilises again in the third inhibitory cycle.
(TIF)

**S4 Fig. The temporal code for three inputs. a |** The image with three inputs. "E" had the highest luminance, followed by "A", and "T". **b |** In absence of inhibition, the three output nodes corresponding to the inputs activate in succession. The amplitude of the unattended "T" is notably reduced compared to "E" and "A". **c |** With a 10Hz inhibition, as used in the previous simulations, "E" is read out as the only letter in the first cycle, "A" activates in the second cycle. Following that, a stable temporal code with "E" and "A" is produced in each cycle. **d |** With a slower inhibition of 6Hz, the three letters activate in the first and second cycle, ordered based on their luminance. However, in the second and forth cycle, the temporal code only consists of the letters "E" and "A", whereby "E" activates twice. In the fifth cycle shown here, "T" activates briefly after "E" is activated.
(TIF)

**S5 Fig. The read-out of two simultaneous stimuli as a function of a phase shift in layer 1, indicated in the title (in ms).** The temporal code is more strongly affected when shifting the

phase of the inhibitory oscillation layer 1 compared to layer 2 (compare Fig 8). Notably, shifting the phase of the inhibition in the first layer mainly seems to affect the activations in the first two cycle of the inhibition (see 16.4–81.8ms), however, the temporal code often appears to recover within three cycles.
(TIF)

## Author Contributions

**Conceptualization:** Katharina Duecker, Marco Idiart, Marcel van Gerven, Ole Jensen.

**Investigation:** Katharina Duecker.

**Methodology:** Katharina Duecker, Marco Idiart, Marcel van Gerven, Ole Jensen.

**Project administration:** Katharina Duecker.

**Supervision:** Marco Idiart, Marcel van Gerven, Ole Jensen.

**Validation:** Katharina Duecker.

**Visualization:** Katharina Duecker.

**Writing – original draft:** Katharina Duecker, Ole Jensen.

**Writing – review & editing:** Katharina Duecker, Marco Idiart, Marcel van Gerven, Ole Jensen.

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
