## [Decision Letter · Decision Letter 0]

5 Jun 2024

Dear Ms Duecker,

Thank you very much for submitting your manuscript "Oscillations in an Artificial Neural Network Convert Competing Inputs into a Temporal Code" for consideration at PLOS Computational Biology.

As with all papers reviewed by the journal, your manuscript was reviewed by members of the editorial board and by several independent reviewers. In light of the reviews (below this email), we would like to invite the resubmission of a revised version that takes into account the reviewers' comments.

We cannot make any decision about publication until we have seen the revised manuscript and your response to the reviewers' comments. Your revised manuscript is also likely to be sent to reviewers for further evaluation.

Sincerely,

Stefano Panzeri

Academic Editor

PLOS Computational Biology

Lyle Graham

Section Editor

PLOS Computational Biology

Dear Authors, thanks for submitting this work to PLoS Computational Biology. As you will see below, the Reviewers found your paper of interest. However, they recommend that several improvements are made. We look forward to receiving the revised version. Yours truly, Stefano Panzeri (Academic Editor of this submission)

Reviewer's Responses to Questions

**Comments to the Authors:**

Reviewer #1: There is a broad range of temporal dynamics in the neural system, and these temporal dynamics are thought to have an important function in the representation of information, especially multiple competing information. In this manuscript, the authors trained the ANN to recognize letters, and with adding of temporal dynamics in the hidden layer, the ANN was able to read out sequentially for two letters presented at the same time. Moreover, the sequence of readout is along the phase of inhibitory oscillations. The results of the study proved that the inhibitory oscillations help to segregate the information in time. Overall, this study is a timely and insight-providing study that demonstrates the positive effects of oscillations on information readout. It was a pleasure to read such a well-designed and well-written study, and I think that both the neuroscience and machine learning fields will have interest in the results of this study, and I would like to see it published.

I have 3 small suggestions that the author might consider.

1. I would like the authors to discuss how the findings of this study contribute to our understanding of how the neural system works.

2. I would like to know if adding other frequencies of oscillations to the hidden layer by changing the time parameter would produce similar results to this study? In other words, could the authors discuss the similarities and differences between brain information processing and ANN information processing, especially in the time scales.

3. There is a small typo, the second line of 2.1 on page 3. Should it be Figure 2a?

Reviewer #2: My comments are in the attached document.

Reviewer #3: The review is uploaded as attachment.

**Have the authors made all data and (if applicable) computational code underlying the findings in their manuscript fully available?**

Reviewer #1: Yes

Reviewer #2: Yes

Reviewer #3: Yes

PLOS authors have the option to publish the peer review history of their article (what does this mean?). If published, this will include your full peer review and any attached files.

Reviewer #1: **Yes: **Jianrong Jia

Reviewer #2: **Yes: **Lou Zonca

Reviewer #3: **Yes: **Simone Blanco Malerba
---

## [Decision Letter · Decision Letter 1]

17 Aug 2024

Dear Ms Duecker,

We are pleased to inform you that your manuscript 'Oscillations in an Artificial Neural Network Convert Competing Inputs into a Temporal Code' has been provisionally accepted for publication in PLOS Computational Biology.

Best regards,

Stefano Panzeri

Academic Editor

PLOS Computational Biology

Lyle Graham

Section Editor

PLOS Computational Biology

Reviewer's Responses to Questions

**Comments to the Authors:**

Reviewer #1: Thanks to the author's efforts to revise the article, the current version is much better than the previous one. I have no further questions.

**Have the authors made all data and (if applicable) computational code underlying the findings in their manuscript fully available?**

Reviewer #1: Yes

PLOS authors have the option to publish the peer review history of their article (what does this mean?). If published, this will include your full peer review and any attached files.

Reviewer #1: **Yes: **Jianrong Jia

---

## [Editor Report · Acceptance letter]

5 Sep 2024

PCOMPBIOL-D-24-00669R1 

Oscillations in an Artificial Neural Network Convert Competing Inputs into a Temporal Code

Dear Dr Duecker,

I am pleased to inform you that your manuscript has been formally accepted for publication in PLOS Computational Biology. Your manuscript is now with our production department and you will be notified of the publication date in due course.

With kind regards,

Anita Estes
